# Position: Assistive Agents Need Accessibility Alignment

**Jie Hu** [1]   **Changyuan Yan** [1]   **Yu Zheng** [1]   **Ziqian Wang** [1]   **Jiaming Zhang** [1]

## Abstract

Assistive agents for Blind and Visually Impaired (BVI) users require accessibility alignment as a first-class design objective. Despite rapid progress in agentic AI, most systems are designed and evaluated under assumptions of sighted interaction, low-cost verification, and tolerable trial-and-error, leading to systematic failures in assistive scenarios that cannot be resolved by model scaling or post-hoc interface adaptations alone. Drawing on an analysis of 778 assistance task instances from prior work, we show that current agentic AI remain prone to failure in assistive scenarios due to mismatches between sighted-user design assumptions and the verification, risk, and interaction constraints faced by BVI users. We argue that accessibility should be treated as an alignment problem rather than a peripheral usability concern. To this end, we introduce accessibility alignment and propose a lifecycle-oriented design pipeline for accessibility-aligned assistive agents, spanning user research, system design, deployment and post-deployment iteration. We conclude that BVI-centered assistive tasks provide a critical stress test for agentic AI and motivate a broader shift toward inclusive agent design.

## 1. Introduction

Recent advances in agentic artificial intelligence (Ferrag et al., 2025; Acharya et al., 2025) have enabled AI systems to function as increasingly general-purpose assistants, integrating multi-step reasoning (Zhu et al., 2025), tool use (Singh et al., 2025), and autonomous decision making (Viswanathan, 2025). These capabilities have motivated growing interest in deploying agents across real-world domains. In accessibility-oriented contexts, agentic AI has been explored as a means of supporting BVI users in navigation (Hwang et al., 2025), information access (Natalie et al., 2025), and everyday task assistance (Oliveira et al., 2022).

However, assistive settings pose requirements that are not captured by the ability to generate plausible or task-relevant responses. The central question is whether users can safely rely on an agent when independent verification is costly, interaction bandwidth is limited, and the consequences of error are asymmetric. This challenge is particularly salient in assistive navigation, which provides high-stakes and empirically observable examples.

StreetReaderAI (Froehlich et al., 2025) improves the accessibility of street-view exploration through context-aware multimodal interaction, yet its evidence is derived from static panoramas whereas real-world environments are dynamic. Consequently, an agent may issue confident guidance about sidewalks, crossings, or landmarks that have changed due to construction, temporary barriers, or traffic conditions, while BVI users may have limited ability to verify such changes in situ. Similarly, a recent study (Chang et al., 2025) of ChatGPT live video chat with blind participants reports that current systems are more reliable in static scenes than in dynamic situations, with inaccurate spatial judgments and hallucinations that may introduce confusion and risk. Even in constrained benchmarks, miscalibration remains evident. Prior work (He et al., 2024) evaluates ChatGPT-4o on short-range wayfinding queries and includes cases in which the appropriate response is to recognize insufficient evidence, yet incorrect guidance remains common. Collectively, these navigation-centered findings highlight a broader principle for assistive agents, namely that the capacity to defer, request additional evidence, or decline to act is as consequential as the capacity to produce an action plan.

These failures cannot be reduced to interface limitations alone. They reflect underlying policy decisions about how an agent solicits evidence, represents and communicates uncertainty, calibrates autonomy, and supports error recovery. In navigation, errors in perception or state estimation can be transformed into fluent instructions, while compressed non-visual output channels may limit the user's ability to diagnose the basis of a recommendation or recover from an erroneous one. Analogous dynamics arise beyond navigation,

---

[1]School of Artificial Intelligence and Robotics, Hunan University, Changsha, China. Correspondence to: Jiaming Zhang <jiamingzhang@hnu.edu.cn>.

*Proceedings of the 43rd International Conference on Machine Learning*, Seoul, South Korea. PMLR 306, 2026. Copyright 2026 by the author(s).

including assistive reading, device operation, and digital workflows such as form completion, purchasing, scheduling, and smart home control. In these settings, hidden state, tool failures, partial observability, and irreversible actions can similarly produce silent failures and overconfident assistance.

The absence of accessibility alignment leads to systematic failures that cannot be addressed solely through model scaling or post-hoc interface adaptations. For BVI users, many assistive tasks involve non-verifiable outputs, safety-critical and asymmetric error costs, heightened cognitive demands, and elevated privacy risks. Under these conditions, errors may remain undetected, recovery may be difficult, and misplaced trust may have serious consequences. BVI-centered assistive scenarios therefore constitute a revealing stress test for agentic AI, exposing design assumptions that may be acceptable for sighted users but fail in accessibility-critical contexts.

These observations motivate our central position that assistive agents require accessibility alignment. We define *accessibility alignment* as the compatibility between the objectives, behaviors, interaction patterns of assistive agents, and the evaluation criteria and the abilities, constraints, and lived experiences of BVI users. Accessibility alignment should not be understood as an automatic consequence of general capability scaling or as a synonym for interface-level accessibility compliance. Rather, it constitutes a distinct alignment objective that determines what an agent optimizes, how it acts under uncertainty, how it communicates with users, and how its performance is evaluated.

To substantiate this position, we conducted a large-scale analysis of assistive tasks extracted from 417 previous works, comprising 778 task instances related to BVI assistance. From these instances, we derive a task-centric taxonomy that characterizes the breadth of real-world assistive needs. Building on this taxonomy, we analyze why current agentic AI systems remain insufficiently aligned with such needs. We argue that the central souce of misalignment is not merely limited model capabilities, but a mismatch between prevailing design assumptions and the verification constraints, error tolerance, autonomy, and interaction conditions of accessibility-critical use. These misalignments manifest across mobility and safety, reading and text access, object centered daily operations, and goal directed visual question answering, indicating that even technically capable agents may remain unreliable in assistive contexts without accessibility-specific alignment.

This paper contributes to this discussion by formulating accessibility alignment as a distinct alignment objective for assistive agents, where success is defined not only by task completion but also by verifiability, risk sensitivity, interaction efficiency, and recoverability under uncertainty. It

further proposes a lifecycle-oriented pipeline for designing, deploying, and iteration accessibility-aligned assistive agents, thereby connecting accessibility requirements to system specification, runtime policy, evaluation metrics and post-deployment feedback.

**Conflict of Interest Disclosure.** The authors declare that there are no financial conflicts of interest related to this work.

## 2. Background

### 2.1. Assistive Technologies for BVI Users

According to the World Health Organization (Organization et al., 2019), at least 2.2 billion people worldwide have near or distance vision impairment. Traditional assistive technologies such as white canes and screen readers (Lazar et al., 2007) have long served as essential mediating tools for accessing physical and digital environments. Recent advances in computer vision have further enabled AI-powered visual assistance systems, including smart belts (Argüello Prada & Santacruz Forero, 2022), electronic guide canes (Khan et al., 2021), intelligent smart glasses (Zheng et al., 2024), and quadrupedal guide robots (Chen et al., 2022b). While these systems demonstrate meaningful progress, their effectiveness in complex real-world settings is often constrained by limited adaptability and robustness. Vision-based aids commonly focus on narrow functions such as obstacle detection, while providing limited support for broader situational awareness. Wearable systems that rely on GPS, IMU or other sensors may struggle with localization errors in signal-degraded environments. Methods built on static scene assumptions can also become brittle under dynamic conditions such as moving crowds, changing obstacles, or unpredictable traffic. These limitations highlight that hardware innovation and perceptual capability alone are insufficient for reliable assistance. Effective support for BVI users requires systems that can reason about user intent, prioritize safety-relevant information, and adapt assistance strategies in real time. This motivates a shift from isolated, single-function devices towards assistive agents that integrate perception, reasoning, and autonomous action in a more holistic and adaptive manner.

### 2.2. Agentic Systems in Accessibility-Critical Settings

Unlike passive models that primarily process information, agentic systems (Sapkota et al., 2025) operate as active entities that perceive, plan, and act within an environment. They coordinate language-based reasoning with tool use while maintaining task state across interactions, enabling multi-step behavior over extended horizons rather than isolated single-turn responses (Fang et al., 2025). For BVI users, these capabilities shift assistance from passive de-

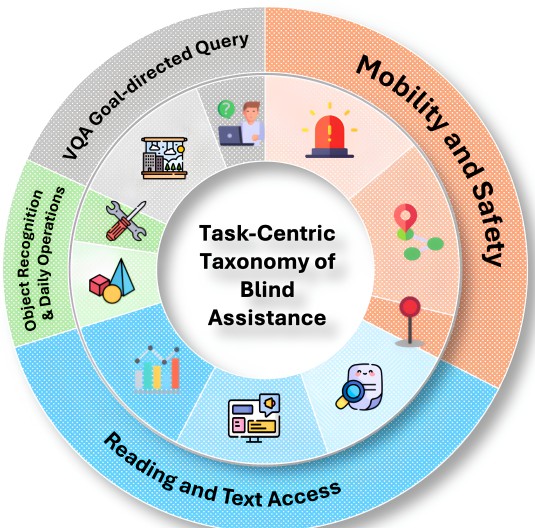

*Figure 1.* **Task-Centric Taxonomy of Blind Assistance and Distribution of Assistive Task Instances.** Distribution of 778 assistive task instances across four domains and their subcategories, highlighting dominant needs in Reading and Text Access (35%) and Mobility and Safety (34%).

scription to goal-directed task support. An assistive agent may, for instance, navigate graphical interfaces to obtain services or guide users through physical spaces rather than merely identify objects or describe scenes. However, this shift also makes accessibility failures more consequential. Mainstream agents are typically trained and evaluated under assumptions that users can verify intermediate outputs, tolerate trial-and-error correction, and recover from mistakes with low cost. In accessibility-critical deployments, these assumptions often break down because BVI users have limited access to visual evidence, constrained interaction bandwidth, and asymmetric exposure to the consequences of error. Agentic systems therefore require accessibility-specific alignment rather than direct transfer of general-purpose agent design.

## 3. Task Taxonomy and Practical Needs of BVI Users

To establish an empirical foundation for assistive agent design, we conducted a systematic literature review covering work from 2012 to 2025. We surveyed 417 publications across Computer Vision (Zhang et al., 2022), Generative AI (Fu et al., 2025), Robotics (Lu et al., 2021), and Human-Computer Interaction (HCI) (Jeanneret Medina et al., 2022). To bridge the gap between abstract system capabilities and concrete user needs, we distilled high-level task descriptions into 778 fine-grained task instances.

Through qualitative coding based on task interdependence and constraint types, we organized these instances into four

primary categories: (1) Mobility and Safety, (2) Reading and Text Access, (3) Object Recognition and Daily Operations, and (4) VQA Goal-directed Query. As shown in Fig. 1, the frequency distribution of these tasks reflects the dominant research emphases in assistive technology while also providing an empirical proxy for recurring, high-priority needs in BVI users' daily lives.

### 3.1. Mobility and Safety

This category covers tasks that support safe locomotion and navigation in physical environments. For BVI users, mobility is central to independent living and often requires continuous, closed-loop interaction with dynamic and partially observable surroundings. Errors in this category are especially consequential, as failures can directly lead to immediate hazards or physical harm.

> **Safety-first autonomy under uncertainty.**

Assistive agents for mobility and safety must support conservative, risk-aware decision-making that accounts for asymmetric and safety-critical error costs. This requires prioritizing safety over efficiency and modulating autonomy according to environmental uncertainty and user context.

**Hazard Perception and Alerts (108 instances).** This task category concerns the perception and communication of potential hazards, including static (Tang et al., 2021) and dynamic (Kuribayashi et al., 2024) obstacles, overhead dangers (Rangam et al., 2025), and broader environmental risks such as traffic or construction zones (Aljarbouh et al., 2024; Zhang et al., 2025). Existing systems have adapted perception techniques to identify hazards and deliver audio (Kim et al., 2025) or haptic (Galapia et al., 2024) alerts. However, work in this area often inherits modeling assumptions from non-assistive settings, which can obscure user-specific costs and misalign design priorities in safety-critical assistance. For BVI users, hazard perception may serve as a primary source of environmental awareness, making both missed hazards and poorly calibrated alerts directly safety-relevant. These properties require accessibility-aware prioritization, uncertainty handling, and alert timing beyond generic perception accuracy.

**Path Planning and Navigation (116 instances).** Path planning (AbuJabal et al., 2024) and navigation (Abidi et al., 2024) involve guiding users toward a destination through safe and efficient routes, often requiring instructions at multiple levels of granularity. These instructions may range from high-level directions (Chen et al., 2022a) such as "turn left after the curb" to fine-grained (Hong et al., 2020), action-level guidance, such as "walk straight for ten meters and turn left". In assistive contexts, navigation systems must adapt to user walking speed, environmental changes, and unexpected obstacles while avoiding excessive instruction

load. Unlike navigation for sighted users or fully embodied robots, assistive navigation cannot assume that route choices and corrections can be visually verified by the end user.

**Localization, Orientation, and Relocation (29 instances).** Localization (Apostolopoulos et al., 2014) and orientation (Fernandes et al., 2019) tasks concern estimating a user's current position and relation to surrounding landmarks or goals, while relocation involves guiding users from the current location to a specific target. These tasks commonly rely on GPS (Drawil et al., 2012), vision-based localization (Adorni et al., 2001), or indoor positioning systems (Wahab et al., 2022). For BVI users, localization errors or delays can propagate into navigation errors with compounding safety implications. Localization in assistive agents should therefore be integrated with navigation and hazard perception rather than treated as an isolated technical module.

### 3.2. Reading and Text Access

This category covers tasks related to accessing, interpreting, and interacting with textual information in physical and digital environments. Reading and text access enable BVI users to engage with documents, interfaces, and everyday informational artifacts, but errors can become consequential when text mediates safety, finance, health or access to services.

> **Verifiable information mediation.**

Assistive agents for reading and text access must align information delivery with high accuracy, explicit uncertainty communication, and calibrated confidence. Accessibility alignment in this category requires treating reading not merely as visual perception, but as trustworthy information mediation.

**General Document Reading (95 instances).** This category includes short-form text (Kuzdeuov et al., 2024), such as labels, signs and menus, as well as longer structured documents (Tang et al., 2025; Huynh & Lin, 2024), such as articles, books, or reports. Assistive systems must support both rapid access to brief information and sustained interaction with extended text, including navigation across sections, headings, and references. Although these tasks differ in scale, they share common accessibility challenges related to accuracy, context preservation, and error detection.

**Interactive Digital Reading and UI Navigation (100 instances).** Interactive digital reading includes navigating web pages (Huynh & Lin, 2024; Moterani & Lin, 2025), emails (Sharevski & Zeidieh, 2024), PDFs (Kumar & Wang, 2024), and application interfaces (Sunkara et al., 2025), often mediated through screen readers. These tasks frequently require searching (Moon et al., 2024), locating (Xu et al.,

2023), and interacting with interface elements such as buttons (Mowar et al., 2024), forms (Schmitt-Koopmann et al., 2025). Unlike passive reading, digital interaction couples text understanding with user actions, increasing the consequences of misinterpretation and the cognitive demands on users.

**Non-linear Visual Documents (98 instances).** Non-linear visual documents include charts (Alcaraz-Martinez et al., 2024), graphs (Srinivasan et al., 2023), and complex tables (Kumar & Wang, 2024) that do not follow a simple linear text structure. Assistive systems must translate these representations into accessible formats that preserve relationships, trends, and spatial organization. Errors or over-simplifications in such translations can fundamentally distort meaning, making faithful abstraction a central accessibility concern..

### 3.3. Object Recognition and Daily Operations

This category focuses on tasks involving understanding, locating, and interacting with objects in everyday environments. Such tasks often require tightly coupling perception and action, since recognition alone is insufficient without reliable guidance for subsequent user behavior.

> **Grounded perception for reliable action.**

Assistive agents for object recognition and daily operations must align perception and action with context-sensitive grounding and reliable action guidance, moving beyond object labeling toward actionable, situation-aware assistance.

**Object Understanding (56 instances).** Object understanding encompasses recognizing objects and interpreting their attributes or states (Do et al., 2025), such as size (Bhowmick & Hazarika, 2017), color (Kuzdeuov et al., 2024), temperature (Mathis & Schöning, 2025), or whether an appliance is on or off (Jiang et al., 2024). For BVI users, misidentifying object properties or states can lead to inappropriate or unsafe actions, highlighting the need for grounded, context-aware interpretation rather than isolated recognition.

**Object-Centered Interaction and Manipulation (35 instances)** These tasks (Jiang et al., 2024; Xu et al., 2023) involve assisting users with household devices such as microwaves, washing machines, or televisions. These tasks require translating perceptual information into step-by-step operational guidance, often under time pressure or safety constraints. Reliability and clarity are therefore critical, as incorrect instructions can result in device misuse or hazards.

### 3.4. VQA Goal-directed Query

This category captures tasks in which users ask goal-oriented questions grounded in visual context, and seek information that directly informs decisions or actions. Un-

like generic visual question answering, these queries are embedded in real-world intent and often require responses that support immediate action.

> **Intent-aware answers that support action.**

Assistive agents for goal-directed visual queries must align responses with user intent, provide action-oriented answers, and preserve contextual continuity across interactions. Purely descriptive responses are insufficient when queries are situated within real-world decision-making.

**Situational Understanding (96 instances).** Situational understanding (Chang et al., 2025) includes describing scenes and interpreting ongoing activities or events. This may involve explaining spatial layouts, identifying dynamic changes, or recognizing behaviors relevant to the user's goals. For BVI users, situational understanding supports anticipatory decision-making rather than passive awareness alone.

**Goal-directed Object Queries (45 instances).** Goal-directed object queries (Bigham et al., 2010; Tseng et al., 2022) focus on locating or identifying specific objects in service of an immediate goal, such as finding an exit or determining the position of a particular item. Effective assistance requires integrating spatial reasoning with actionable guidance rather than merely producing factual answers.

# 4. Why Assistive Agents Fail Without Accessibility Alignment

Current agentic AI often fails in assistive settings because its objectives, policies, and interaction assumptions are not aligned with accessibility-grounded constraints. We structure the diagnosis by first identifying the environmental stressors that characterize assistive tasks for BVI users, then analyzing the recurring failure modes those stressors produce, and finally explaining why generic capabilities such as planning or tool use are insufficient to address them. We subsequently define accessibility alignment as a four-dimensional framework that directly responds to these failures.

## 4.1. The Stressors That Define BVI Assistive Settings

Assistive scenarios operate under constraints that violate the standard assumptions of general-purpose agent development. We identify four such stressors.

**Limited verifiability.** Users frequently cannot independently verify agent outputs. When an agent describes a crosswalk as clear, a blind user has no immediate means to confirm that assessment before acting. Under limited verifiability, overconfident completion becomes hazardous because users may unknowingly rely on false or incomplete information.

information.

**High-cost errors.** Error costs in assistive contexts are often severe and irreversible. While a hallucination in a low-stakes creative task may be benign, a hallucination in mobility guidance can cause physical injury, a misreading of a medication label can cause medical harm, and a mistake in a financial interaction can cause material loss. Assistive agents must therefore minimize worst-case risk rather than optimizing only for average-case helpfulness.

**Cognitive burden.** BVI users commonly rely on audio or haptic feedback while simultaneously navigating physical spaces or maintaining situational awareness. Verbose or poorly structured output imposes cognitive load during safety-critical tasks, forcing users to filter, parse, and reconstruct spatial information under real-time demands.

**Privacy exposure.** Assistance interactions routinely capture sensitive data, such as views of private interiors, medical records, and bystanders in public spaces. An agent optimized primarily for convenience may over-collect or vocalize such information without explicit user consent. Privacy exposure is therefore not peripheral to accessibility but integral to the risk structure of assistive deployment.

## 4.2. Recurring Failure Modes in Assistive Agents

Under the four stressors identified above, unaligned agents exhibit recurring failure patterns that are especially consequential in accessibility-critical use. We identify four such patterns and trace each to its driving stressor combination.

**Silent failures.** The agent acts on incorrect information without exposing uncertainty or provenance. In goal-directed visual question answering, confident but wrong answers can mask hallucinations and leave users unaware of the risk. This pattern is driven primarily by limited verifiability and asymmetric error costs, because the user cannot cheaply detect discrepancies and the cost of undetected errors is severe.

**Overconfident hallucinations.** The agent generates plausible but incorrect content to fill evidential gaps. When reading a blurred prescription label or interpreting an ambiguous crossing, plausibility becomes a safety hazard because the user lacks independent means of verification. This pattern is a direct consequence of limited verifiability interacting with an agent design that prioritizes fluent completion over uncertainty signaling.

**Miscalibrated autonomy.** The agent fails to adjust its decision authority to the risk and confidence of the situation. It may book a ride to an unverified location without confirmation or, conversely, demand unnecessary clarifications in low-risk tasks. This pattern stems from asymmetric error costs and constrained bandwidth, which together limit the

user's capacity to supervise every agent decision.

**Interaction-induced cognitive overload.** The agent delivers verbose, non-linear, or poorly timed information that demands excessive attention. In physically situated tasks, such failures distract users from real-world hazards and directly undermine safety. This pattern expresses constrained interaction bandwidth under real-time task demands.

These four modes reinforce one another. Silent failures and hallucinations erode trust, which increases cognitive burden as users attempt to mentally verify every output. Mis-calibrated autonomy amplifies this effect by obstructing verification when risk is highest and by consuming bandwidth with unnecessary interaction when risk is low. Addressing accessibility failures therefore requires a unified framework that targets all four patterns simultaneously.

### 4.3. Origins of Misalignment: Assumptions and Capability Gaps

The failure modes diagnosed above arise from two layers of structural misalignment between current agent design practices and the requirements of assistive deployment.

**Implicit design assumptions.** Current agentic systems are typically developed under three assumptions that do not transfer to accessibility-critical contexts. The verification assumption presumes that users can rapidly audit agent outputs, often through visual inspection. For BVI users, such auditing is frequently infeasible, and its absence directly enables silent failures and overconfident hallucinations. The error-tolerance assumption presumes that mistakes are visible and recoverable through iterative correction. In mobility or medication tasks, however, users cannot safely test a dangerous route or an uncertain dosage to determine correctness, which leaves mis-calibrated autonomy unchecked. The shared-context assumption presumes that users and agents perceive the same visual environment. In practice, BVI users depend on the agent to construct and communicate that context, creating an information asymmetry that agents are rarely trained to bridge and that contributes to interaction-induced cognitive overload.

**Capability-need mismatch.** Even when these assumptions are made explicit, general agent capabilities do not natively accommodate accessibility-specific requirements. Path planning can optimize for efficiency but lacks interfaces for encoding conservative risk preferences, such as avoiding unlit crossings or favoring routes with tactile pavement. Vision tools can return labels and confidence scores yet often lack abstention mechanisms that would prevent hallucinations from being communicated authoritatively. Memory modules can store user preferences for personalization but, without privacy-aware constraints, may also retain sensitive contextual histories in ways that undermine trust. In each

case, the capability exists in principle, yet the architecture may provide no explicit mechanism for incorporating the safety margins, abstention policies, and privacy controls that assistive tasks demand. Accessibility failures therefore reflect not only insufficient capability but also a structural gap between what agent architectures expose and what assistive contexts require.

### 4.4. A Four-dimensional Framework of Accessibility Alignment

We define accessibility alignment along four dimensions, each addressing a specific requirement that current agentic systems underspecify. In assistive settings, alignment is not limited to task completion. It also requires satisfying accessibility-defined success criteria under safety constraints, non-visual interaction, and persistent uncertainty.

**Goal alignment.** This dimension addresses the gap between generic task completion and accessibility-defined success. Reaching a destination does not constitute success if the resulting route involves unsafe crossings or offers no recovery opportunity after navigation errors. Document reading is not successful if critical fields such as dosage or contraindications are inaccurate or unsupported by sufficient evidence. Goal alignment requires explicit safety margins, reliability guarantees for critical information, and error recovery procedures under uncertainty.

**Interaction alignment.** Failures in interaction design often stem from paradigms that assume visual context and high-bandwidth feedback. For BVI users, effective assistance requires concise, structured, and immediately actionable communication. It further demands confirmation strategies compatible with non-visual modalities, including chunked instructions, explicit landmark references, and robust correction loops that enable efficient error detection and recovery. Interaction alignment is especially critical in multi-turn assistance and device operation.

**Risk alignment.** This dimension addresses failures under uncertainty in safety-critical and privacy-sensitive contexts. Many agentic systems are optimized to maximize task success or produce coherent outputs rather than to adopt conservative policies that account for asymmetric risk. In assistive settings, uncertainty should systematically trigger safety-oriented behavior, including requesting additional context, proposing safer alternatives, pausing, or declining to act. Risk alignment also requires privacy-respecting behavior, with data minimization by default and explicit communication of privacy implications whenever sensing or sharing may occur.

**Lifecycle alignment.** Sustaining long-term trust requires preventing silent performance degradation over time. Assistive agents operate over extended periods in dynamic en-

vironments, alongside evolving user routines and changing device configurations. Without monitoring and feedback, errors accumulate unnoticed, leading to brittle behavior and eroded user confidence. Lifecycle alignment therefore demands logging and auditing mechanisms, closed-loop user feedback, and update processes that prioritize safety, stability, and accountability.

These four dimensions form a coupled system rather than independent modules. Goal alignment defines what counts as high-risk, interaction alignment specifies how confirmations operate under non-visual constraints, risk alignment enforces conservative behavior when uncertainty is elevated, and lifecycle alignment ensures that these properties remain stable across deployment. The dimensions jointly respond to the four stressors and, taken together, constitute the operational definition of accessibility alignment for assistive agents.

# 5. An Accessibility-Aligned Pipeline for Assistive Agents

To operationalize the four alignment dimensions, we propose a lifecycle-oriented pipeline that translates accessibility constraints into concrete specifications, runtime policies, evaluation targets, and post-deployment update mechanisms. The pipeline embeds risk management, non-visual interaction, and user trust across three phases: design, deployment, and post-deployment iteration.

### 5.1. Phase I: Design for Alignment

The design phase converts an assistive need into an implementable specification through six artifacts.

**Task Card.** Task anchoring situates the target functionality within the taxonomy and defines operational boundaries, including environment dynamics, user constraints, and explicitly disallowed conditions.

**Accessibility Success Specification.** Success criteria are defined in accessibility terms beyond generic task completion. They specify safety margins, reliability and provenance expectations for critical information, and explicit error recovery procedures.

**Interaction Contract.** Interaction is formalized as a low-bandwidth, non-visual protocol. It specifies how instructions are chunked, how landmarks and orientation cues are expressed, when confirmations are required, and how correction loops operate.

**Risk and Uncertainty Policy.** Observable uncertainty signals are bound to conservative actions, such as requesting additional context, proposing safer alternatives, pausing, or declining to proceed.

**Privacy Manifest.** Data minimization rules and user-facing authorization requirements make sensing, retention, and sharing implications explicit.

**Autonomy Calibration Specification.** Autonomy is defined as a dynamic variable controlled by risk, confidence, and user context. Explicit user overrides and safe fallback behaviors are specified.

Together, these artifacts define what constitutes success, how the agent communicates, and when it must adopt a more conservative stance.

### 5.2. Phase II: Deployment

Deployment translates design artifacts into enforceable runtime behavior and establishes a controlled operational boundary.

Pre-deployment validation centers on the unacceptable failure set defined in the *Accessibility Success Specification*. Stress tests target safety-critical, uncertainty-heavy, and privacy-sensitive scenarios. Validation further verifies that the *Interaction Contract* remains usable under non-visual constraints.

At runtime, guardrails enforce risk-triggered downgrades in autonomy, safe pause and escalation pathways, and user-accessible controls for immediate intervention. Onboarding materials communicate capability boundaries and interaction expectations, including how confirmations work, how to correct mistakes, and how privacy settings affect sensing and logging. Deployment succeeds when conservative behavior is enforced rather than left to discretionary model behavior, and when users can reliably predict agent behavior under uncertainty.

### 5.3. Phase III: Post-deployment Iteration

The iteration phase prevents failures from accumulating silently across evolving environments, routines, and device configurations.

The system maintains a minimal, privacy-respecting logging and feedback plan that captures near-misses, uncertainty triggers, and user corrections without excessive reporting burden. Observed issues are triaged by mapping incidents to specific stressors and failure modes, then diagnosing which alignment dimension is underspecified or violated. Updates follow a safe update protocol with regression tests that target red-line scenarios, interaction contract compliance, and risk-policy consistency. Rollback mechanisms are required when updates introduce safety regressions. Over time, controlled personalization is enabled through explicit user settings and transparent adaptation rules, while conservative defaults are preserved.

This three-phase structure provides a direct pathway from

*Table 1.* Operationalizing accessibility alignment through navigation and medication-reading case studies.

| Case | Red-line failures | Uncertainty triggers | Evaluation shift | Runtime implications |
|---|---|---|---|---|
| Navigation assistance for safe mobility | *Decisive crossing or route instructions when localization, curb geometry, or obstacle status cannot be reliably assessed.* | **Localization drift**, occlusion, ambiguous map or **landmark** evidence, dynamic **obstacles**, poor sensing conditions, inconsistent **traffic** cues. | From {**SPL, path length, and travel time**} to {**unsafe instruction rate, risk-trigger compliance, recovery success**, instruction latency, and confidence calibration}. | Uncertainty estimation, conservative route selection, **autonomy downgrade**, landmark-based **non-visual** instructions, **safe pause**, escalation to human assistance under high uncertainty. |
| Medication label and leaflet reading | *Confidently reporting unsupported dosage, frequency, contraindications, adverse effects, interactions, or unit conversions from partial or ambiguous evidence.* | Blur, occlusion, folded or curved packages, dense typography, **conflicting OCR** or VLM candidates, **low confidence** on numeric fields, layout ambiguity. | From {**OCR accuracy, CER/WER**, and answer accuracy} to {**critical-field accuracy, critical hallucination rate, abstention precision and recall**, and **recapture success**}. | Field-level confidence, ambiguity detection, structured output templates, recapture policy, **critical-field verification**, **abstention**, escalation to pharmacist when key information is unverifiable. |

observed failures to revised specifications, deployment mechanisms, and continual improvement.

## 5.4. Operationalizing the Pipeline Through Two Case Studies

We instantiate the pipeline in two representative high-stakes scenarios: navigation assistance and medication reading. The following table contrasts the unaligned baseline with the accessibility-aligned design across four operational dimensions. Table 1 summarizes how the two cases translate accessibility alignment into red-line failures, uncertainty triggers, evaluation shifts, and runtime implications.

**Case 1: Navigation assistance for safe mobility.** The *Task Card* specifies outdoor navigation in dynamic traffic and explicitly disallows decisive crossing instructions at uncontrolled intersections when critical signals cannot be reliably assessed. The *Accessibility Success Specification* encodes safety margins, alternative-route requirements, and recovery procedures for localization drift. The *Interaction Contract* mandates landmark-based, chunked instructions with mandatory confirmations at high-risk points and immediate stop controls. The *Risk and Uncertainty Policy* enumerates signals such as localization instability and occlusion, binding them to pausing, requesting additional cues, or proposing safer detours. During deployment, validation stresses red-line scenarios including crossings and intersection handling, and onboarding clarifies capability boundaries and override mechanisms. Post-deployment, near-misses and user corrections are mapped back to stressors and alignment gaps, with regression tests preserving conservative behavior on safety-critical maneuvers.

**Case 2: Reading medication leaflets and labels.** The *Task Card* defines likely inputs such as folded leaflets and bottle labels under variable lighting and dense typography. The *Accessibility Success Specification* elevates dosage, frequency, contraindications, and interactions as critical fields requiring reliability signaling and explicit recovery when extraction is ambiguous. The *Interaction Contract* enforces a structured output format supporting field navigation and confirmation of high-stakes values. The *Risk and Uncertainty Policy* treats blur, occlusion, and conflicting field candidates as triggers for recapture, alternative presentation, pausing, or escalation to a pharmacist. Deployment validation covers hard layouts, low-light conditions, and numeric-unit edge cases, ensuring the system never collapses ambiguity into definitive statements. Post-deployment, user corrections and recapture patterns are triaged by stressor and alignment dimension, and regression suites maintain conservative behavior for all critical fields.

Across both cases, three shared patterns emerge from the alignment pipeline. The first pattern is that evaluation shifts from task-completion metrics to safety- and verifiability-aware metrics, each directly encoding a stressor from Sec. 4.1. A second, related pattern concerns where uncertainty is represented in the architecture. In the navigation case, localization uncertainty triggers route downgrading and a pause; in the medication case, field-level confidence triggers recapture or escalation to a pharmacist. In both settings, uncertainty is surfaced at the decision point rather than buried in upstream processing. Together, these patterns point to a third, overarching principle. Conservative behavior is not an optional post-hoc adjustment but a property enforced through runtime guardrails and escalation pathways, making safety the default operating mode. These patterns generalize across assistive domains and constitute the operational core of accessibility alignment for assistive agents.

# 6. Alternative Views

## 6.1. Generalization vs. Accessibility

> **"General-purpose agents will eventually solve this."**

A common counterargument holds that accessibility challenges for BVI users will resolve naturally as agentic AI systems become more general, capable, and robust. Under this view, advances in reasoning, perception, and planning driven by larger models and broader training data will eventually cover assistive needs without requiring dedicated design.

We argue that this position conflates generalization with accessibility. Generalization concerns an agent's ability to perform well across tasks and environments, typically under benchmarked or average conditions. Accessibility concerns whether an agent's goals, behaviors, and interactions fit the abilities, constraints, and risk profiles of specific user populations. An agent can generalize across tasks while remaining unreliable for users with disabilities (Chang et al., 2025; He et al., 2024). Crucially, increased autonomy and confidence, often treated as markers of progress in general-purpose agents, can worsen accessibility outcomes when users cannot independently detect errors or recover from them (Froehlich et al., 2025). Our analysis of 778 task instances suggests that many failures in BVI assistive scenarios do not arise from a lack of task competence but from misaligned assumptions about verification, error tolerance, interaction pacing, and autonomy that persist even when agents exhibit strong general-purpose performance.

Accessibility is therefore neither a special case of generalization nor a property that reliably emerges from scale (Ferrag et al., 2025). It constitutes a distinct alignment objective that must be addressed explicitly in agent design, evaluation, and deployment. Treating accessibility as a downstream problem to be solved later risks systematically excluding users whose needs do not match the assumptions embedded in current agentic systems.

## 6.2. HCI vs. AI

> **"It is not an AI, but a HCI problem."**

Another frequent response is that accessibility challenges should be handled primarily through human-computer interaction rather than AI system design. Under this framing, the difficulties BVI users face are treated as interface limitations, solvable through improved input modalities, better screen reader support, or refined interaction flows, with the underlying agent architecture left unchanged (Lazar et al., 2007; Jeanneret Medina et al., 2022).

While HCI plays a crucial role in accessibility, this framing becomes inadequate in the context of agentic AI. Unlike traditional interactive systems, agentic AI systems are not passive tools that merely respond to user input. They are decision-making entities that plan actions and operate autonomously over extended time horizons (Sapkota et al., 2025; Fang et al., 2025). As a result, accessibility constraints shape not only how users interact with agents but also how agents should reason, act, and manage uncertainty.

Many of the failures identified in our analysis originate upstream of the interface. These include inappropriate autonomy, miscalibrated confidence, unsafe action selection, and a failure to account for asymmetric error costs (Chang et al., 2025). Such failures cannot be corrected solely through interface adaptations because they reflect misalignment in agent goals, decision policies, and evaluation criteria. We therefore argue that accessibility for assistive agents is not only an HCI concern but a foundational AI design problem. Addressing it requires integrating accessibility requirements into agent architectures, planning and control mechanisms, interaction strategies, and post-deployment learning. Treating accessibility as an interface-layer issue encourages post-hoc fixes at the expense of establishing accessibility as a core requirement for responsible agentic AI.

## 6.3. Limitations

Our taxonomy is derived from published task instances and may underrepresent real-world practices of BVI users outside research settings. The proposed pipeline remains a design framework rather than an implementation specification, defining lifecycle stages and design artifacts while leaving architectural choices and system trade-offs to future work. Future work should validate the framework through longitudinal deployments, quantitative measures of trust and uncertainty calibration, and standardized benchmarks reflecting the constraints documented in our taxonomy.

# 7. Conclusion

In this position paper, we grounded our claim through a large-scale analysis of 417 prior works and 778 assistive task cases, producing a task-centric taxonomy of blind assistance and surfacing recurring failure patterns that persist despite strong general-purpose capabilities. We proposed accessibility alignment as a practical framing for agent design and evaluation, and outlined a lifecycle-oriented pipeline to operationalize it. We hope this perspective encourages the community to build assistive agents that are not only more capable, but also more accessible for the BVI users.

**Acknowledgment.** This work was supported by National Natural Science Foundation of China under Grant No. 62503166.

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
