# OpenReview forum: "Position: Assistive Agents Need Accessibility Alignment"
_ICML.cc/2026/Position_Paper_Track — ICML 2026 Position Paper Track spotlight_

### Official Review · Reviewer_ikB4 · 2026-03-10

**Significance:** 3
**Argument Clarity:** 3
**Rating:** 5
**Confidence:** 3

**Questions:**

1. The lifecycle pipeline is one of the paper’s main contributions. Could the authors clarify which parts they see as the most immediately actionable for current agentic systems, and which require longer-term methodological development?
2. The paper argues that accessibility alignment should shape evaluation as well as design. What concrete benchmarks or metrics do the authors think would best capture reliability, uncertainty calibration, and safe autonomy in BVI assistive settings?

**Alternative Views Section:**

Yes

**Compliance With Llm Reviewing Policy A Conservative:**

Affirmed.

**Discussion Potential:**

4

**Final Justification:**

The paper makes a clear argument that accessibility for assistive agents for blind and visually impaired users should be treated as an alignment problem rather than a post-hoc interface issue. Its large-scale review, task taxonomy, failure analysis, and lifecycle pipeline make the position concrete and well supported. The rebuttal addressed my questions clearly and strengthened my confidence in the paper’s framing and contribution. I therefore maintain my Accept recommendation.

**Paper Summary:**

This paper argues that accessibility design must be the primary design goal for assistive agents for blind and visually impaired users. This article does not treat accessibility design as an ex-post interface problem, but rather as a coordination problem of agents, involving goals, behaviors, interaction patterns, and evaluation criteria. To support this view, the authors analyze 417 existing literature and 778 auxiliary task examples, construct a task-centric classification system for blind auxiliary agents, identify common failure modes and design assumptions flaws, and propose a lifecycle-oriented pipeline for designing, deploying, and iterating accessibility-aligned assistive agents.

**Position:**

Yes

**Position In Title:**

Yes

**Related Work:**

4

**Strengths And Weaknesses:**

This is a strong and well-structured position paper. The central claim is clear, important, and consistently argued throughout the paper: accessibility for assistive agents should be treated as an alignment problem rather than a peripheral usability concern. The paper is stronger than a purely opinion-based position piece because it grounds its argument in a large-scale analysis of prior work and develops a task-centric taxonomy spanning mobility and safety, reading and text access, object recognition and daily operations, and goal-directed visual queries. I also found the failure analysis compelling: the discussion of limited verifiability, asymmetric error costs, cognitive burden, privacy exposure, and recurring failure modes gives the position real substance. The proposed lifecycle pipeline is another strength, since it makes the paper constructive rather than purely critical.

The main limitation is that some parts of the proposal remain high-level. While the lifecycle pipeline is useful and concrete for a position paper, it is still a design framework rather than an implementation or evaluation protocol, and some operational details are left open. In addition, parts of the paper’s overall message build on broader user-centered design and accessibility principles that may already be familiar in adjacent HCI and accessibility communities, so the novelty lies more in synthesizing and reframing these ideas for agentic AI than in introducing a completely new perspective.

**Support:**

4

---

> ### Author Rebuttal · Authors · 2026-03-31
>
> > **W1+Q2: The proposals remain high-level. The pipeline is useful but not an implementation or evaluation protocol.**
>
> We acknowledge that proposals remain high-level because our position represents a completely new perspective. In the development of assistive agents, considering their accessibility alignment is essential. Therefore, we hope this position will inspire community researchers to refine more concrete training and evaluation methods for benchmarking assistive agents, including but not limited to user-centric study, trainig objective, agent architecture, and agent evaluation.
>
> The pipeline is intentionally defined at a more abstract level because our goal is to propose a framework that remains applicable across different task types, model choices, and agent architectures, rather than narrowing down the pipeline to one specific implementation protocal. A more concrete formulation that is directly bound to a particular VLM, RL setup, or embodied stack would narrow the contribution and weaken the generality of the position paper. Proposing more detailed or concrete implementation or evaluation protocals would be left for our future work and the whole community.
>
> Nonetheless, to address your concern, we have carefully added some cases in the **task-specific table** (Table R1 for Reviewer `2KGt`) to illustrate how the proposed high-level framework can be instantiated differently across major BVI assistive task categories.                           |
>
> > **W2: User-centered design and accessibility principles may be familiar in HCI, so the novelty lies more in synthesizing and reframing these ideas for agentic AI than in introducing a completely new perspective.**
>
> Our work is conducted from total 778 task instances from 417 papers related to blind and low vision assistanc. We also found that many of them are familiar in the HCI and accessibility communities. Our contribution and novelty, however, is not simply to restate those principles, but to show how they must be carefully reframed as an alignment problem in developing assistive agents. In our work, accessibility alignment concerns the agent’s goals, behaviors, interaction patterns, and evaluation criteria, not only input and output accessibility or screen-reader compatibility.
>
> More importantly, we hope this position can encourage the AI and HCI communities to develop more concrete alignment methods for assistive agents, involving user-centric study, training objective, agent architecture, and agent evaluation.
>
> > **Q1: The lifecycle pipeline is one of the paper’s main contributions. Could the authors clarify which parts they see as the most immediately actionable for current agentic systems, and which require longer-term methodological development?**
>
> In our view, the most immediately actionable parts of the pipeline are accessibility success specification and interaction alignment. For example, in current agentic systems, developers can already redefine success from generic task completion to accessibility specific success, specify unacceptable failure sets, and validate systems on high risk scenarios before deployment. In practice, this means adapting or auditing current agents against accessibility specific success criteria so that they are evaluated against accessibility relevant criteria such as unsafe crossing instruction rate, critical field hallucination rate in medication reading, or unsafe action suggestion rate in daily operations, rather than generic helpfulness alone.
>
> By contrast, the longer term directions are lifecycle alignment and controlled personalization. These require more mature deployment infrastructure for continuous logging, failure triage, rollback, and trust maintenance over time, as well as methods for personalization that preserve conservative defaults. We therefore see these as important but less immediately deployable, since they depend on stronger monitoring, feedback, and benchmarking ecosystems than most current assistive agents yet support.
>
> We hope our response has addressed your concerns and look forward to further discussion.

---

> > ### Author Rebuttal · Reviewer_ikB4 · 2026-04-04
> >
> > My concerns have been well addressed, so I will keep my original positive rating.

---

### Official Review · Reviewer_gVk9 · 2026-03-12

**Significance:** 3
**Argument Clarity:** 3
**Rating:** 5
**Confidence:** 4

**Questions:**

See the strengths and weaknesses part.

**Alternative Views Section:**

Yes

**Compliance With Llm Reviewing Policy A Conservative:**

Affirmed.

**Discussion Potential:**

3

**Paper Summary:**

This paper argues that current agent systems fail Blind and Visually Impaired (BVI) users because they rely on implicit design assumptions suited for sighted individuals, such as low-cost visual verification and high error tolerance. Based on an analysis of 778 real-world assistive task instances, the authors demonstrate that simply scaling general AI capabilities does not resolve these safety-critical gaps. Instead, they introduce Accessibility Alignment, proposing that assistive agents must explicitly align their goals, interaction styles, and risk policies with BVI constraints. Furthermore, they outline a lifecycle pipeline spanning design, deployment, and post-deployment iteration to build safer, more reliable assistive agents.

**Position:**

Yes

**Position In Title:**

Yes

**Related Work:**

3

**Strengths And Weaknesses:**

Strength
===

1. The paper tackles a highly relevant and socially impactful issue within the AI4Good space. Addressing the specific needs of BVI users in the era of agentic AI is a critical endeavor, especially since mainstream AI progress often relies on assumptions that implicitly center sighted users.
2. The authors provide a robust, empirically grounded taxonomy derived from 778 real-world assistive task instances across four main domains (Mobility and Safety, Reading and Text Access, Object Recognition and Daily Operations, and VQA Goal-directed Query). Furthermore, clearly articulating the unique environmental stressors (such as limited verifiability, asymmetric high-cost errors, and cognitive burden) successfully highlights how assistive contexts fundamentally deviate from general-purpose agent assumptions.
3. The paper excels in diagnosing why current agentic AI fails in BVI scenarios, specifically calling out issues like silent failures, overconfident hallucinations, and mis-calibrated autonomy. By framing these failures as a fundamental alignment problem rather than mere capability limitations or simple HCI/interface gaps, the authors provide a strong conceptual foundation that will guide future research.

Weakness
===

1. While the proposed three-phase accessibility-aligned pipeline is conceptually sound, it remains somewhat high-level and theoretical. The paper would benefit from deeper, practical insights or a technical case study demonstrating exactly how to operationalize these architectural choices and manage system trade-offs in practice.
2. Evaluating whether an agent has achieved true accessibility alignment is a critical challenge that feels under-addressed. This paper lacks concrete discussion on the specific quantitative and qualitative metrics required to measure this. The authors briefly note the future need for standardized benchmarks and quantitative measures of trust, but expanding on what these evaluation frameworks might actually look like would substantially strengthen the paper's immediate contribution.

**Support:**

3

---

> ### Author Rebuttal · Authors · 2026-03-31
>
> > **W1: The three-phase pipeline sound too high-level, and lack practical or technical demonstration of how to operationalize the architecture and handle system trade-offs in practice.**
>
> We acknowledge that the current pipeline is somehow high-level. Because the position is completely new in this field, and our goal is to propose an accessibility alignment discipline that can generalize across tasks, models, and system stacks, rather than to prescribe one very specific implementation. If we tied the framework directly to a specific agent, a particular RL setup, or one embodied model, that would narrow down the discusion of accessible alignment. The manuscript therefore concludes the pipeline as a design framework, while leaving concrete choices and tradeoffs to future system-specific work in the community.
>
> On the other hand, Sec. 5 already aims to operationalize it through concrete design artifacts and two case examples, showing how accessibility requirements can be translated into runtime guardrails, conservative fallback behaviors, and post deployment regression tests. In the revision, we will make this motivation more explicit and strengthen the practical side by adding more cases.
>
> > **W2: Evaluating whether an agent has achieved true accessibility alignment is a critical challenge that feels under-addressed. This paper lacks concrete discussion on the specific quantitative and qualitative metrics required to measure this. The authors briefly note the future need for standardized benchmarks and quantitative measures of trust, but expanding on what these evaluation frameworks might actually look like would substantially strengthen the paper's immediate contribution.**
>
> We agree that evaluation is a critical part of accessibility alignment, and that the current version only briefly points to this need. To address this more concretely, we will add a task-specific discussion of training objectives, evaluation metrics, and architectural implications for different assistive scenarios grounded in our taxonomy. The same **task-specific table** (Table R1 for Reviewer `2KGt`) is intended to make these evaluation directions more explicit, including how evaluation priorities differ across mobility, reading, daily operations, and goal-directed VQA.
>
> We hope our response has addressed your concerns and look forward to further discussion.

---

### Official Review · Reviewer_G1bj · 2026-03-12

**Significance:** 3
**Argument Clarity:** 2
**Rating:** 4
**Confidence:** 4

**Questions:**

1. Why is the argument framed specifically around agentic AI systems rather than AI systems more broadly?

2. How does accessibility alignment differ from existing work on safety, verification, and uncertainty management in AI systems?

3. How does the proposed pipeline differ from existing application level deployment practices for assistive AI systems?

4. Could the authors outline concrete research questions or methodological directions that would help address the challenges discussed?

**Alternative Views Section:**

Yes

**Compliance With Llm Reviewing Policy A Conservative:**

Affirmed.

**Discussion Potential:**

3

**Final Justification:**

Overall, the problem addressed is meaningful and relevant, although the level of novelty is somewhat limited. However, it remains valuable in fostering discussion and further emphasizing this perspective. I encourage the authors to strengthen the conceptual contribution by developing a clearer methodological framework, better distinguishing it from broader AI concerns, and focusing more explicitly on generalization and concrete research directions.

**Paper Summary:**

The paper argues that agentic AI systems should be designed with accessibility alignment in mind, particularly for users who are blind or visually impaired. The authors suggest that assistive agents must consider user abilities, constraints, and lived experiences when determining goals, behaviors, interaction patterns, and evaluation criteria. The paper highlights challenges arising when AI systems operate in real-world assistive scenarios where verification may be difficult and the consequences of errors are significant. Through examples such as navigation tasks, the authors emphasize that assistive agents should not only generate actions but also determine when to defer decisions, seek additional information, or decline to act. The paper frames accessibility alignment as a design principle intended to ensure that assistive AI systems behave safely and reliably in high-stakes contexts.

**Position:**

Yes

**Position In Title:**

Yes

**Related Work:**

3

**Strengths And Weaknesses:**

**Strengths:**

- The paper highlights an important and socially relevant problem related to accessibility and assistive AI systems.
- The position advocating accessibility-aware design for assistive agents is clearly articulated.
- The paper raises useful considerations about how accessibility constraints influence agent interaction and behavior.

**Weaknesses:**

- The scope of the paper appears narrow and focuses primarily on assistive applications rather than broader machine learning challenges.
- Many of the issues discussed, such as safety, verification, and uncertainty handling, also apply to AI systems more generally and are not clearly distinguished in the accessibility context.
- The proposed pipeline resembles standard application deployment practices and lacks methodological novelty.
- The paper lacks formalism or a conceptual framework describing how accessibility aligned agents should be developed or trained.
- The paper does not clearly outline concrete research questions or methodological directions for addressing the identified challenges.

**Detailed Review:**

The paper advocates designing agentic AI systems that are aligned with the needs and constraints of users with visual impairments. The motivation is well grounded, and the discussion highlights real challenges that arise when AI systems are deployed in assistive contexts where errors may have significant consequences. The emphasis on agents that can determine when to act, defer decisions, or request better evidence is an important design consideration for assistive technologies.

However, the paper largely frames the issue as an application level challenge rather than a machine learning research problem. Many of the limitations described appear to stem from current system capabilities and deployment practices rather than from fundamental algorithmic limitations. The paper does not clearly explain why these concerns apply specifically to agentic AI rather than to AI systems more broadly.
In addition, the connection to machine learning methodology is limited. The discussion focuses on design considerations and deployment concerns but does not present a clear conceptual framework for how accessibility aligned systems should be modeled, trained, or evaluated.

The proposed pipeline resembles standard application design and does not introduce new computational approaches or learning methods. As a result, the paper remains at a high level and does not provide clear research directions or methodological proposals.

Overall, the topic is important and worthy of discussion, particularly from an application and societal perspective. However, the contribution would be stronger if it articulated clearer distinctions from existing safety and verification challenges in AI systems and provided more concrete research directions or conceptual frameworks relevant to machine learning.

**Support:**

2

---

> ### Author Rebuttal · Authors · 2026-03-31
>
> We thank the reviewer for the valuable comments and suggetions.
> > **W1: The scope of the paper appears narrow.**
>
> Our work and proposals are grounded in 778 task instances from 417 blind and low-vision (BVI) assistance papers, and fundamentally focus on agentic systems integrating perception, reasoning, tool use, uncertainty handling, and autonomous decision-making. We center on assistive settings because they uniquely amplify these core challenges: in BVI contexts, low verifiability and high error costs sharply expose miscalibration, silent failure, and overconfident action. Thus, we frame assistive scenarios not as a narrow application, but as a critical stress test for broader questions in alignment, evaluation, and agent design.
>
> > **W2 + Q2: How accessibility alignment is substantively distinct from broader AI work on safety, verification, and uncertainty management?**
>
> We agree that safety, verification, and uncertainty matter for AI more broadly. Our position is that, in accessibility settings, they should be defined from the user’s perspective rather than only the agent’s. What distinguishes assistive settings is the combination of limited verifiability, asymmetric error costs, cognitive burden, and privacy exposure. In these contexts, safety concerns user safety during situated use, verification concerns whether outputs remain actionable when independent confirmation is difficult, and uncertainty handling should trigger conservative behavior rather than fluent completion.
>
> > **W3 + Q3: Lack methodological novelty. How the proposed pipeline differs from existing application-level deployment practices?**
>
> Our novelty lies not in the generic sequence of design, deployment, and iteration, but in redefining each stage around **accessibility alignment** for assistive agents. We introduce accessibility-specific artifacts such as the Task Card, Interaction Contract, Risk and Uncertainty Policy, and Autonomy Calibration Specification. Unlike standard application-level deployment, uncertainty here is treated as a policy signal that should trigger conservative behavior rather than fluent continuation under ambiguity. We also formalize non-visual interaction and require post-deployment logging, triage, regression testing, and rollback.
>
> > **W4 + W5 + Q4: Conceptual framework for developing accessibility-aligned agents and concrete research questions or methodological directions to address the challenges discussed.**
>
> We thank the reviewer for this constructive comment. Different assistive tasks require tailored operationalization: navigation, medication-reading, and object-manipulation agents need task-specific training objectives and metrics to achieve accessibility alignment. We define our pipeline at a high level not to prescribe a single training objective or benchmark, but to highlight universal accessibility-alignment principles across assistive agents. To address your questions, we added the **task-specific table** (see Table R1 for Reviewer `2KGt`) to make this operationalization more concrete.
>
> Our core research question is accessibility alignment: how to design, deploy, and update assistive agents to ensure their goals, behaviors, and interaction patterns align with BVI users’ abilities, constraints, and risk profiles. Our proposed pipeline serves as the methodological direction to address this question. In revision, we will clarify that the paper’s contribution is both to define accessibility alignment as the core research question and to present the pipeline as a methodological suggestion.
>
> > **Q1: Why is the argument framed specifically around agentic AI systems rather than AI systems more broadly?**
>
> We frame our argument around agentic AI because accessibility challenges become far more consequential when systems shift from **passive information** support to **active task assistance**. As detailed in Sec. 2.2, agentic systems integrate perception, planning, tool use, and long-horizon stateful decision-making. In assistive settings, this means failures are no longer limited to incorrect descriptions, but can lead to erroneous actions, unsafe guidance, or miscalibrated autonomy during task execution.
>
> Our argument is not restricted to narrow agentic AI: it also applies to earlier assistive AI systems (e.g., navigation or reading tools) that shape user decisions under low verifiability and high error costs. We prioritize agentic AI because it represents the field’s dominant development direction, and the shift from passive to highly autonomous assistance makes the alignment problem far more acute and urgent.
>
> In short, while accessibility alignment matters for all AI systems, agentic systems are the clearest, most demanding use case given their sustained action, statefulness, and temporal control. This is why assistive agents deserve focused attention as the field shifts from passive tools to active assistance.
>
> We hope our response addresses your concerns, and we welcome further discussion.

---

> > ### Author Rebuttal · Reviewer_G1bj · 2026-04-03
> >
> > Thank you to the authors for the response and for the clarifications provided in the rebuttal.
> >
> > Overall, the problem addressed is meaningful and relevant, although the level of novelty is somewhat limited. However, it can still be valuable in fostering discussion and further emphasizing this perspective. I encourage the authors to strengthen the conceptual contribution by developing a clearer methodological framework, better distinguishing it from broader AI concerns, and focusing more explicitly on generalization and concrete research directions.
> >
> > With these considerations, I will increase my score from 2 to 4.

---

### Official Review · Reviewer_2KGt · 2026-03-13

**Significance:** 3
**Argument Clarity:** 3
**Rating:** 4
**Confidence:** 3

**Questions:**

Do the authors envision specific benchmark datasets or evaluation frameworks for accessibility-aligned agents?

**Alternative Views Section:**

Yes

**Compliance With Llm Reviewing Policy A Conservative:**

Affirmed.

**Discussion Potential:**

3

**Paper Summary:**

This position paper argues that assistive agents intended for blind and visually impaired (BVI) users must treat accessibility as an alignment goal rather than as an afterthought or only an HCI problem. The authors present an empirical grounding (a literature review of 417 prior works producing 778 assistive task instances) and derive a task-centric taxonomy and a lifecycle-oriented design pipeline to operationalize what they call accessibility alignment.

**Position:**

Yes

**Position In Title:**

Yes

**Related Work:**

3

**Strengths And Weaknesses:**

This paper contains sevaral strengths:

- The paper addresses a critical and underexplored issue: how agentic AI systems interact with accessibility-critical user groups. As AI agents are increasingly deployed in real-world environments, ensuring that they behave safely for users who cannot easily verify outputs is a meaningful and socially impactful research direction.
- The taxonomy of 778 task instances across four domains provides a helpful synthesis of the assistive technology literature. Even though it is derived from existing papers rather than new datasets, it provides a useful organizing lens for researchers interested in accessibility-driven agent design.
- The four failure modes—particularly silent failures and miscalibrated autonomy—capture real risks in assistive scenarios. The discussion effectively highlights how standard AI assumptions (e.g., easy verification, low-cost errors) break down in accessibility contexts.

However, there are also some weaknesses:

- Many of this work’s proposals remain high-level. For example, the accessibility alignment dimensions are conceptually clear but not operationalized in technical terms. Also, it is unclear how these ideas translate into model training objectives, evaluation metrics, or agent architectures.

Overall, this paper raises an important issue about the interaction between agentic AI systems and accessibility-critical users. The concept of accessibility alignment is meaningful and the taxonomy provides a useful perspective on assistive AI tasks. However, the contribution is largely conceptual, and the paper would benefit from stronger methodological rigor and more concrete links to machine learning research. Taking all these into consideration, I recommand a boarderline accept.

**Support:**

3

---

> ### Author Rebuttal · Authors · 2026-03-31
>
> > **W1: Proposals remain high-level and have not yet been translated into concrete technical definitions.**
>
> We acknowledge that our position represents a new perspective and that some proposals remain high-level. In the development of assistive agents, considering accessibility alignment is essential. We hope this position will encourage the community to develop more concrete methods for training and evaluation, including user-centric study, training objective, agent architecture, and agent evaluation.
>
> Besides, accessibility alignment does not take the same form across assistive tasks. Our taxonomy (Sec. 3) shows that different assistive domains imply unique alignment requirements.
>
> Nonetheless, thanks to your suggestions, we have organized concrete BLV assistive cases to compare the use of accessibility alignment, as presented in the following table.
>
> **Table R1: Task-specific case studies of accessibility alignment.**
> | **Task category**  | **BLV assistive scenario**         | **Without accessibility alignment** | **With accessibility alignment** | **Evaluation difference**  | **Architecture difference** |
> |-|-|-|-|-|-|
> | **Mobility and Safety**   | *Guiding a blind user through a street crossing or unfamiliar intersection*  | Prioritizes _route completion_ and fluent guidance, even under uncertainty | Prioritizes **safe, recoverable mobility** with **conservative behavior** under uncertainty                           | From {**SR**, **SPL**, **Path length**, **Completion time**} to **unsafe instruction rate**, **risk-trigger compliance**, **recovery success**, instruction latency, and confidence calibration   | From a generic navigation agent to one with **uncertainty estimation**, **autonomy downgrade**, and landmark-based interaction  |
> | **Reading and Text Access** | *Reading medication labels or folded medication leaflets* | Prioritizes _coverage_ and fluent completion from partial evidence  | Prioritizes **verifiable mediation** of **safety-critical information**, with abstention or recapture under ambiguity | From {**OCR accuracy**, **CER/WER**, answer accuracy, and coverag} to **critical-field accuracy**, **critical hallucination rate**, **abstention precision/recall**, and **unit consistency** | From a generic OCR or VLM pipeline to one with **field-level confidence**, **ambiguity detection**, and escalation for unverifiable content |
> | **Object Recognition and Daily Operations** | *Assisting a blind user in operating a microwave or identifying whether a stove is on* | Treats recognition as sufficient and may produce plausible but unsafe next steps | Prioritizes **safe action support** by jointly modeling perception confidence and action consequences  | From {**classification accuracy**, **F1**, and task completion rate} to **unsafe action suggestion rate**, **actionable state accuracy**, **hazard-free completion**, and correction burden    | From a recognition-first pipeline to a **perception-action architecture** with state verification and safety checks |
> | **Goal-directed VQA**  | *Helping a blind user find an exit, empty seat, or service counter*    | Prioritizes descriptive answer quality or generic VQA correctness    | Prioritizes **concise, action-oriented, spatially grounded guidance** for immediate task completion                   | From {**VQA accuracy**, **BLEU/ROUGE/CIDEr**, and response fluency} to **goal completion rate**, **spatial-action correctness**, **clarification turns**, and **context continuity**   | From a generic VQA system to an **intent-aware planner** with spatial memory and concise non-visual interaction policy   |
>
> > **Q1: Envision benchmark for accessibility-aligned agents?**
>
> We appreciate this question. After our proposed position is accepted by the community, building concrete benchmarks and evaluations will be the following step. Although we know this will be a huge effort and a significant challenge. Our taxonomy of total 778 assistive task instances shows that evaluation should be organized by task domain rather than by generic agent success alone. Besides, it should reflect the accessibility-specific stressors, including limited verifiability, asymmetric error costs, cognitive burden, and privacy exposure.
>
> Furthermore, we suggest accessibility-aligned evaluation as lifecycle-based rather than as a single static benchmark. More specificly, it should combine task-specific benchmarks with pre-deployment stress testing on unacceptable failure sets, runtime monitoring of uncertainty-triggered behaviors and user corrections, and post-deployment regression testing for safety-critical scenarios. Some cases are presented in the table above according to the four major categories.

---

> > ### Author Rebuttal · Reviewer_2KGt · 2026-04-02
> >
> > My questions have been solved, and I maintain my positive score.

---

### Decision · Program_Chairs · 2026-04-30

**Decision:**

Accept (spotlight)

**Comment:**

All reviewers rated the paper positively (2 borderline accepts and 2 accepts).

This is a strong and well-structured position paper. The central claim is clear, important, and consistently argued throughout the paper: accessibility for assistive agents should be treated as an alignment problem rather than a peripheral usability concern. The paper is stronger than a purely opinion-based position piece because it grounds its argument in a large-scale analysis of prior work and develops a task-centric taxonomy spanning mobility and safety, reading and text access, object recognition and daily operations, and goal-directed visual queries. The paper addresses a critical and underexplored issue: how agentic AI systems interact with accessibility-critical user groups.

While the topic is important and worthy of discussion, particularly from an application and societal perspective, the contribution would be stronger if it articulated clearer distinctions from existing safety and verification challenges in AI systems and provided more concrete research directions or conceptual frameworks relevant to machine learning.